# PuzzleJax: a benchmark for reasoning and learning

**annonymous**

## Abstract

We introduce *PuzzleJAX*, a GPU-accelerated puzzle game engine and description language designed to support rapid benchmarking of tree search, reinforcement learning, and LLM reasoning abilities. Unlike existing GPU-accelerated learning environments that provide hard-coded implementations of fixed sets of games, *PuzzleJAX* allows dynamic compilation of any game expressible in its domain-specific language (DSL). This DSL follows *PuzzleScript*, which is a popular and accessible online game engine for designing puzzle games. In this paper, we validate in *PuzzleJAX* several hundred of the thousands of games designed in *PuzzleScript* by both professional designers and casual creators since its release in 2013, thereby demonstrating *PuzzleJAX*'s coverage of an expansive, expressive, and human-relevant space of tasks. By analyzing the performance of search, learning, and language models on these games, we show that *PuzzleJAX* can naturally express tasks that are both simple and intuitive to understand, yet often deeply challenging to master, requiring a combination of control, planning, and high-level insight.[1]

## 1 Introduction

Games, including board game, card games, and various types of video games, have been used to train and test AI methods for a long time. The beauty of this is that depending on the particular game, and how it is represented to the AI system, it can test different AI capabilities. This includes learning, planning, and reasoning; specialized game-based benchmarks have been developed for different methods, such as tree search, reinforcement learning, and large language models [31].

Relative to other genres (e.g. strategy games, platforming games, arcade games), *puzzle games* have received comparatively less research attention. These games are typically single-player, with full or nearly full state observability and relatively modest action spaces. What puzzle games lack in dexterity-based challenges, they make up for in tests of logical inference and long-horizon planning. Puzzle games also range from simple representation (e.g. *Sokoban*, *Boulder Dash*, or *Lemmings*) to expansive and complex (e.g. *Portal*, *The Witness*, or *Baba is You*). We argue that even simple tile-based puzzle games represent an important unsolved frontier in game AI research and help test increasingly important aspects of artificial "cognition" in the era of large language models.

Rather than isolating a single puzzle game or group of games as a target or benchmark, we propose a framework for analyzing and evaluating tile-based puzzle games more generally. Our approach builds on *PuzzleScript*, a domain-specific language for expressing 2D tile-based puzzle games already used by game developers around the world. We reimplement the core functionalities of *PuzzleScript* in JAX, a modern Python library for hardware-accelerated code. The end result is a benchmark of over 400 diverse game environments and the capacity to generate and automatically compile completely novel rulesets. Our benchmark, *PuzzleJAX*, avoids the common problem of model overfitting by offering a vast array of environment dynamics and objectives while still providing a unified observation and action space. *PuzzleJAX* is completely interoperable with existing *PuzzleScript* game descriptions, giving easy access to thousands of unique and human-validated game environments. *PuzzleJAX* is

---

[1]Our code is available at `https://anonymous.4open.science/r/script-doctor-BDA4`

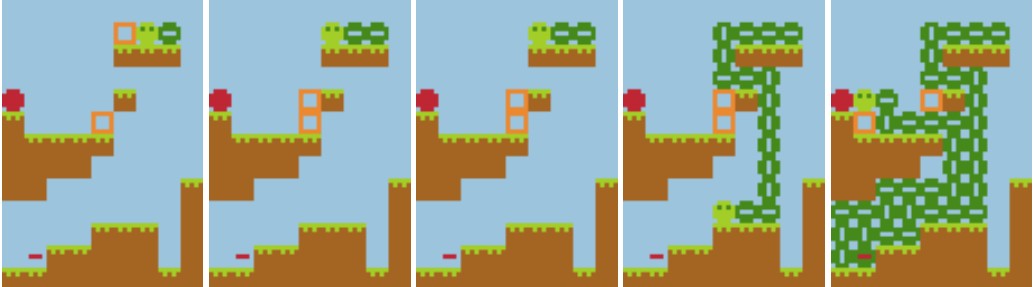

(a) In **Lime Rick**, the player controls a caterpillar creature whose head can rise vertically by at most 3 tiles. The player must navigate the level, using their own body and pushable crates to reach the exit against gravity.

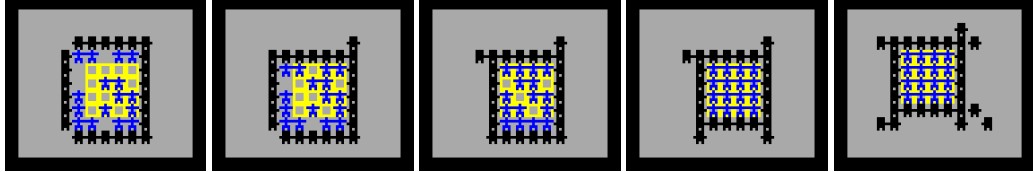

(b) In **Kettle**, the player controls multiple walls of policemen, which can each move in one direction, and must strategically sequence moves to push (or "kettle") a group of civilians into a compact, confined square.

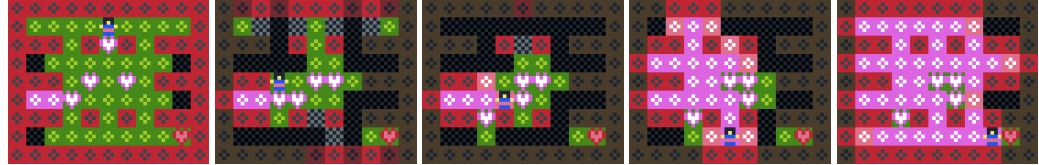

(c) In **Take Heart Lass**, the player must reach the exit (red heart) before they are blocked by the spreadable despair (black tiles). They can push pink hearts to block the despair or unblock hope (pink tiles) that spread and consume despair.

Figure 1: Example games from the framework that showcase the diversity of *PuzzleScript* games.

also fast: by leveraging the power of modern computing hardware, we achieve speed-ups in all the tested games ranging from 300% to 500% compared to existing implementations in JavaScript.

In the following sections, we describe the *PuzzleJAX* language and implementation in detail, provide comparisons to the existing *PuzzleScript* implementation, and showcase initial examples of planning algorithms, reinforcement learning, and LLM-based players interacting with puzzle game environments.

## 2 Related work

Games have been a test bed for AI algorithms, especially Reinforcement Learning algorithms [25], for many years. The reason behind this is the complexity a game offers to an AI algorithm, which can help in benchmarking planning, reasoning, and learning. For example, AlphaGO [24] was an agent that learned to play Go, which defeated the world champion in the game. Similarly, AlphaStar [29] defeated professional StarCraft 2 players, a game known to be one of the most challenging real-time strategy games, and OpenAI Five [4] defeated professional Dota 2 players. Hence, games have been stepping stones for researchers to bring progress to the AI algorithms. To this extent, previous works have seen many games turning into AI benchmarks. Arcade Learning Environment [3] uses Atari games as a benchmark for learning algorithms. Minecraft [6], a popular 3D open-world game, has been used as a benchmark for planning and learning in RL agents [2, 18]. Super Mario Bros have been used as a learning environment as well [8, 11, 20].

Furthermore, generalisation in game-playing RL algorithms has been a core interest among RL researchers. The General Video Game AI (GVGAI) [22] research effort leveraged the Video Game Description Language (VGDL) [7] a Domain Specific Language (DSL) designed to support a large set of arcade-style games, and studied the problem of generalization in RL [28, 10, 19]. Similarly,

NetHack Learning Environment [12] (a port of NetHack) and Crafter [9] (a 2D version of Minecraft) were developed to benchmark generalisation in RL algorithms. *PuzzleJAX*, follows in this line of work, supporting hundreds of existing human games while also providing a DSL that is capable of expressing a diverse range of game mechanics.

Due to the long training time for RL, previous works utilized JAX (a GPU-accelerated language) to speed up the learning process of an agent. JAX is mostly used to implement problems outside of games such as Kinetix [15], a physics-based environment for control tasks. Due to the complexity of game mechanics and rules, fewer frameworks exist in JAX. Craftax [14] (Crafter [9]) and XLand-minigrid [17] (XLand [27] in a minigrid [5]) are two of the game benchmarks ported to JAX. To the best of our knowledge, *PuzzleJAX* is the first JAX-compatible DSL for games.

Lastly, *PuzzleJAX* will also be used to benchmark planning and reasoning abilities in Large Language Models (LLMs) and Vision Language Models (VLMs). Previously, GameTraversalBenchmark [16] created a procedurally generated 2D games where LLMs were benchmarked for planning abilities by traversing the maps. SmartPlay [30] introduced a benchmark for LLMs to play 6 games, including Minecraft and Crafter. Dsgbench [26] introduced 6 strategic games to assess decision-making abilities in LLMs in the benchmark. Similarly, Balrog [21] introduces a benchmark consisting of 6 learning environments, including Crafter and NetHack Learning Environment, for testing agentic capabilities of long-context LLMs and VLMs.

# 3  *PuzzleScript*

*PuzzleScript*, released in 2013 by indie game developer Stephen Lavelle, is a description language and game engine for puzzle games. It is implemented in JavaScript and served on a public website, including an IDE, a debugger, and an interactive player. The central feature of the *PuzzleScript* description language is its *rewrite rules*. The mechanics of the classic box-pushing game Sokoban [1], for example, are defined by the following rule:

```
[ > Player | Crate ] -> [ > Player | > Crate ]
```

This indicates that whenever a Player object is in a cell adjacent to a Crate, and moves toward the Crate, then the Crate likewise moves in this same direction. In general, these rewrite rules describe how spatial patterns of objects and forces distributed over a given game level transform from one timestep to the next.

*PuzzleScript* games are comprised of a single file, which is broken down into eight sections describing different elements of the game:

The **Prelude** section includes metadata such as title, author name, website, and certain global parameters, like whether rules should "tick" at the beginning of an episode of gameplay, or whether the play window should display the entire map or an sub-section of the map centered at the Player.

The **Objects** section defines entities—like the Player and Crate above—that may exist in the game level and interact with one another via rewrite rules. Each object is given a name, an optional single-ASCII-character (for later use in levels), and an optional sprite representation.

The **Legend** section can be used to compositionally define meta-objects which can later be referred to in rules. For example, one might define both Player and Crate as Moveable by stating *Moveable = Player or Crate*, indicating *either* of the component sub-objects is present in a cell. Similarly, the user can define joint-objects that can later be used to indicate the presence of *both* objects simultaneously.

The **Sounds** section defines sound effects that can occur under various conditions, though we ignore it, given that sound effects in *PuzzleScript* games are largely auxiliary.

The **Collision Layers** section lists groups of objects (atomic, joint-, or meta-objects) on separate lines to indicate that these objects collide with one another and therefore cannot overlap.

The **Rules** section defines the mechanics of the game. It includes the left-right pattern rewrite rules like the "player pushes crate" rule described above. It may also prepend these rules with keywords that define, for example, whether they only apply under certain rotations. Rule suffixes may also indicate whether their application triggers a win state, a restart state (e.g. when the player walks into lava), or the repeat application of the overall tick function after the current pass. Within rules, objects (atomic, meta- or joint-objects) may be modified by relative or absolute force indicators

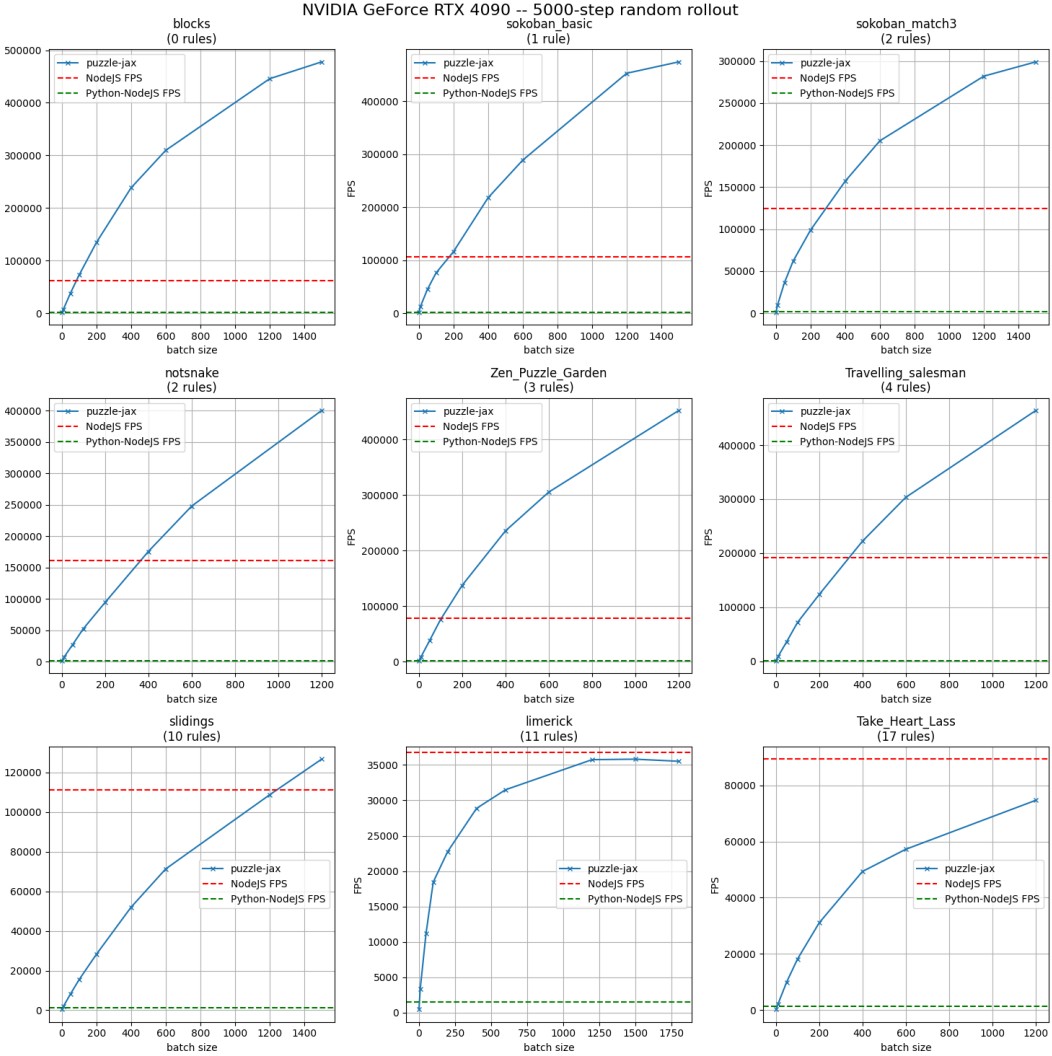

Figure 2: Speed of *PuzzleJAX* compared against a random agent in the original *PuzzleScript* engine, where random actions are carried out internally (NodeJS) or sent from Python (Python-NodeJS).

112  ("$<, >, \wedge, \vee$" and "left, right, up, down" respectively) or other prefixes to indicate e.g. whether an
113  object is stationary or absent from a given cell. Left and right rule patterns may detect or project
114  overlapping objects, respectively, though the same number of cells must be included in left and right
115  patterns. The rules are applied in order from top to bottom and will be repeated by the system until
116  no more matching is happening.

117  The **Win Conditions** section describes a set of necessary conditions which, when satisfied, result in
118  the player "winning" the level. These conditions take the form: "All ObjectA on ObjectB", "Some
119  ObjectA on ObjectB", "No ObjectA", or "Some ObjectA", indicating that all or at least one (some)
120  of a given object (atomic, meta-, or joint-object) must be overlapping with another object type, or
121  that none or at least one (some) of a given object type is present in the level.

122  Finally, the **Levels** section defines the game levels' initial layouts, using a rectangular arrangement
123  of ASCII shorthands for atomic or joint objects. This section may also define natural text messages
124  to be displayed to the player between levels, normally used by designers to convey to the player
125  instructions or narrative elements in the game.

| Game | Solved Levels % | # Total Levels | Max Search Iterations |
|---|---|---|---|
| *Sokoban Basic* | **100%** | 2 | 900 |
| *Sokoban Match3* | **100%** | 2 | 1,1620 |
| *Limerick* | 40% | 10 | 1,000,000 |
| *Blocks* | **100%** | 1 | 788,146 |
| *Slidings* | **100%** | 11 | 12,189 |
| *Notsnake* | 0% | 1 | 42,000 |
| *Traveling Salesman* | **100%** | 11 | 2,204 |
| *Zen Puzzle Garden* | 0% | 5 | 1,000,000 |
| *Multi-Word Dictionary Game* | **100%** | 1 | 15,875 |
| *Take Heart Lass* | 91.6% | 12 | 1,000,000 |
| *Kettle* | **100%** | 11 | 36298 |
| *Constellationz* | **100%** | 5 | 193 |

Table 1: Efficacy of breadth-first search on various *PuzzleScript* games. For each game, we report the percentage of solved levels within 1 million iterations or 1 minute of breadth-first search (out of the total number of levels) as well as the maximum number of search iterations reached in any level.

# 4 *PuzzleJAX* Framework

*PuzzleJAX* is a port of *PuzzleScript* to JAX. The primary goal of the *PuzzleJAX* framework is *fidelity*: to faithfully replicate the *PuzzleScript* engine, unifying a rich, widely-used, and challenging domain with cutting-edge advances in hardware acceleration. We therefore focus on covering as much of *PuzzleScript*'s feature space as possible, carefully validating implemented games and mechanics against their JavaScript counterparts to ensure identical behavior (see subsection 4.1). We emphasize that *PuzzleJAX* is *fully interoperable* with *PuzzleScript*– users and game designers can write novel games with their existing workflows and seamlessly compile them into JAX learning environments without any modification. Our second goal is *speed*: we aim to provide state-of-the-art throughput on a wide range of novel learning environments. *PuzzleScript* is actually a natural candidate for hardware acceleration on modern GPUs, as games are formulated entirely in terms of *local rewrite rules* that modify the tile-based game state and can be applied simultaneously over the entire board. Finally, our third goal is *accessibility*. We provide interpretable environment code, readable syntax, and support for a wide variety of search algorithms, learning frameworks, and reasoning models.

## 4.1 Implementing *PuzzleJAX*

*PuzzleScript* game description files can be cast as a context-free grammar [13]. We define such a grammar in Lark [23], and use it to transform *PuzzleScript* game description files into structured Python objects. Levels are represented as multihot binary arrays, with channels representing the presence of atomic objects and the directional movement or action forces that can be applied to each object (with an additional channel indicating cells affected by the player's last action).

To apply rewrite rules, we effectively detect the presence of objects and forces in the left pattern by applying a convolution to the level, then project the right pattern by passing the resulting array of binary activations through a transposed convolution. For rules involving meta-objects or ambiguous forces (via the "moving" keyword), we apply custom detection and projection functions to convolutional patches of the level, identifying the extant atomic objects or forces at runtime. Alternatively, one might expand such abstract rules to a set of atomic sub-rules; the effect of such a decision on run- and compile-time given variously compositionally complex rule and object definitions could be explored in future work.

Rules in *PuzzleScript* allow for matching the left side to all the possible locations in the level, which could be more than one. In general, if all of the distinct input kernels comprising a left pattern are present at one or more points in a level, then the rule application function attempts to apply all output kernels in the right pattern at whatever points their left-pattern counterparts are active. This is implemented in a JITted jax *while* loop over active indices. If any of these kernel projection operations change the level array, then the rule has been applied.

Generally, rules defined in *PuzzleScript* files are broken down at compile time into a *Rule Group* comprising 4 rotated variants (or 2 given the rule prefixes "vertical" or "horizontal"; or 1 given the rule prefixes "left", "right", "up", or "down"). Each rule in a group is applied sequentially as many times as possible until it no longer has an effect on the level state. Similarly, each rule group is applied until it has no effect before moving on to the next. The game file may also manually define looping rule blocks by enclosing rule definitions in "startLoop" and "endLoop" lines, in which case the enclosed sequence of rule groups is repeatedly executed until ineffective. Finally, a movement rule is likewise applied until it has no effect, which rule attempts to move objects one tile in the direction of any force assigned to them (and if so, removing the force), attempting to apply such forces as they appear in scan-order in the level, and to objects in the order they are defined in the game's collision layers section.

This hierarchical rule execution sequence can be leveraged to create complex dynamics between ticks of the engine, such as gravity moving an object down. *PuzzleJAX* replicates this rule execution logic with a series of nested JAX while loops. Wherever possible, we place logic inside python for loop over static variables (i.e., the number of blocks, groups within each block, and rules within each group). This comes at a cost in terms of compile time (as JAX effectively "unrolls" for loop iterations into distinct blocks of compiled XLA code). Alternatively, we can use JAX *switch* to select from among the list of all rule functions. We found that using the switch significantly affects runtime speed, so we decided to go with increasing compilation time, given that our target is deep learning algorithms with high sample complexity.

## 4.2 *PuzzleJAX* games

We tailor a small dataset of sample games, which are mechanically simple and often challenging, and which, taken together, give a sense of the breadth of the space of possible games supported by *PuzzleJAX*. We describe some of them here and in Figure 1.

**Blocks** is the simplest game with no rules; the game is mainly in the level design where the player needs to navigate a maze to reach the exit.

**Sokoban** is the canonical *PuzzleScript* game, based on the game of the same title, in which the player must navigate a top-down grid of traversible and wall tiles, pushing crates onto targets. The challenge is to sequence moves such that crates do not wind up "deadlocked" in a position (e.g. a corner) from which they cannot be moved onto a target tile.

**Sokoban Match 3**: as above, but when the player arranges 3 or more crates in a horizontal or vertical line, they disappear (as Match-3 games like *Candy Crush*). The goal is to make sure that all the crates disappear from the level.

In **Multi-word Dictionary**, the player arranges English letters by either pushing or pulling in different directions to spell a correct English word.

**Travelling salesman** involves a player on a graph of nodes projected onto the map grid, with varying connectivity patterns (represented by edges connecting the border of two nodes). The player must produce a path that covers all of these nodes from their starting position. The player colors nodes once it traverses them, is unable to return to colored nodes, and wins once all nodes have been colored.

**Zen Puzzle Garden**, similar to the previous game, allows the player to "rake" (similar to coloring the tile) each cell in a central square of sand without retracing its steps, while at the same time avoiding increasingly complex arrangements of obstacles within the sand patch.

**NotSnake** also follows the same idea of coloring cells. The player swaps the color of tiles as it moves, with the aim of coloring the entire level, but is able to retrace its steps with the consequence of flipping these tiles back to their original color.

In **Slidings**, the player can control any one of a number of boulders (swapping between them by pressing the Action key), which they can "slide" in any direction until they hit an obstacle. The player must arrange these boulders onto targets in a fixed number of moves.

In **Constellationz**, the player controls a group of objects simultaneously, all of which must be moved onto targets (without any target left unoccupied); when player objects move onto special teleportation/cloning cells, they disappear, and all unoccupied instances of these cloning cells spawn new player objects (this game uses multi-kernel/non-local patterns to implement this mechanic).

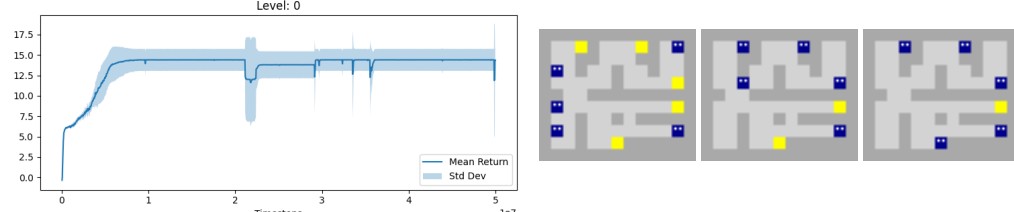

Figure 3: In *Blocks*, a PPO Reinforcement Learning agent quickly learns to improve score according to the heuristic, but falls into a sub-optimal strategy in which one of the Player blocks is trapped in a dead-end corridor adjacent to the one containing the last remaining target.

In **Lime Rick** shown in Figure 1a, the player controls a caterpillar creature whose head can rise vertically by at most 3 tiles. The player must navigate the level, using their own body and pushable crates to reach the exit against gravity. Gravity affects the player's movement and pushable blocks.

In **Kettle** shown in Figure 1b, the player controls multiple walls of policemen, which can each move in one direction, and must strategically sequence moves to push (or "kettle") a group of civilians into a compact, confined square.

In **Take Heart Lass** shown in Figure 1c, the player must reach the exit (red heart) before they are blocked by the spreadable despair (black tiles). They can push pink hearts to block the despair or unblock hope (pink tiles) that spread and consume despair.

In **Atlas Shrank**, the player is a platformer puzzle game where the player needs to reach the exit. The player can't jump, but it can move horizontally, vertically, and diagonally (if stair-shaped solids exist). Most levels have boulders that the player can carry and place in another place to create a ladder to help them navigate the complex level space.

# 5 Results

We converted *PuzzleScript* into a standalone NodeJS package that could be called from Python without a browser, removing GUI-related functionality for rendering text, images, and sounds (we call that Nodejs framework). This framework will be our baseline of comparison for all the following experiments. All the experiments were conducted on the same consumer machine with an NVIDIA GeForce RTX 4090 GPU and Intel Core i9-1100K @ 3.5 GHz CPU.

## 5.1 Speed profiling

To compare the original *PuzzleScript* engine with *PuzzleJAX*, we measured frames per second for a random agent taking random actions. We have two types of random agents, one completely in Nodejs and another one where the actions are taken in Python and sent to Nodejs framework, which better represents the RL training scenario that *PuzzleJAX* targets.

In Figure 2, we plot the number of frames per second obtained by *PuzzleJAX* on the first level of various *PuzzleScript* games at different batch sizes (i.e. number of environments simulated in parallel). We see that *PuzzleJAX* achieves significant speedups over the original *PuzzleScript* engine given small rule-sets, particularly when integrating the engine with a Python wrapper. The speedup is particularly pronounced at large batch sizes, owing to JAX's efficient vectorization scheme. We note that for games with particularly large numbers of rules (e.g. *Slidings*, *Limerick*, and *Take Heart Lass*), random rollouts conducted within the original *PuzzleScript* engine outperform *PuzzleJAX* (indeed, parallelization via multithreading of the original engine may widen this gap). However, *PuzzleJAX* still handily outpaces the original engine when it is forced to communicate with a Python interface. In the context of modern AI methods that involve training large neural networks or fine-tuning large pre-trained models, it is this scenario that is most relevant. Additionally, training such agents or networks with *PuzzleJAX* would not incur any communication costs between the CPU and GPU because the entire environment is hardware accelerated—a fact which would further hamper pipelines relying on the original engine.

## 5.2 Tree search

To probe the complexity of *PuzzleScript* games, we perform breadth-first search over game states for a small set of games and each of their levels. We limit the search to either 1 million environment steps or 1 minute of elapsed time and report the number of levels solved as well as the maximum number of search iterations reached over all levels in Table 1. We note that the performance of tree search is very "all-or-nothing" as games tend to either be simple enough mechanically that brute force suffices (e.g. *Sokoban* or *Slidings*), or complex enough that even the simplest levels are too difficult to solve (e.g. *Notsnake* or *Zen Puzzle Garden*). In addition, we find that the number of search steps required in a game tends to increase as levels progress, mirroring the increasing levels of planning and problem-solving required of human players.

## 5.3 Reinforcement learning

We train standard PPO on individual levels from our set of example games, parameterizing agents as simple convolutional and fully connected feedforward networks, feeding them the multihot encoded level state as observation, and providing the difference between the distance-to-win heuristics derived from the game's win conditions as reward. This heuristic tries to minimize the distance between player and objects required in winning condition and between objects in the winning condition.

We find that agents quickly learn to generate increased reward, but that this learning almost always converges to incorrect solutions Figure 3. *Sokoban* and *Sokoban Match 3*, while solvable via brute-force search, challenge RL agents that greedily maximize rewards but end up in deadlock states (e.g., pushing boxes to blocked targets). In *LimeRick*, agents may lead players vertically toward the Apple but fall into pits, causing deadlocks. Interestingly, these same games can be quickly brute-force by naive breadth-first tree search.

## 5.4 LLM agents

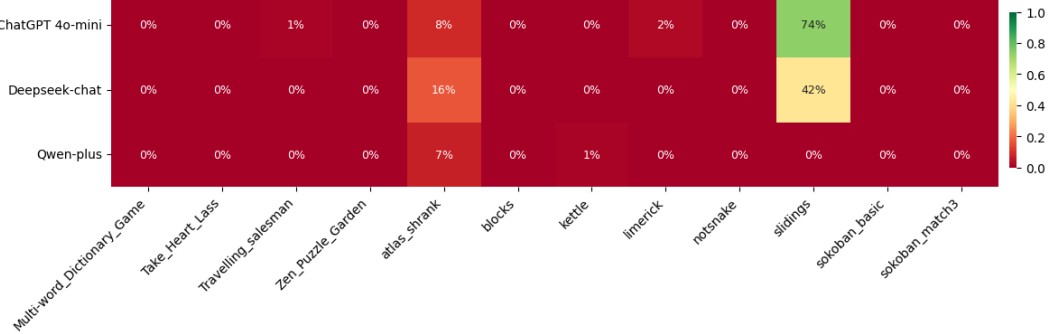

Figure 4: Average Win Rate of three LLMs across 12 games.

In the *PuzzleJAX* benchmark, LLM player agents operate within a structured information framework designed to enable effective puzzle solving without requiring visual interpretation capabilities. The framework provides agents with an `ascii_state` containing both the current game state and a dynamic mapping, complemented by its `rules`, alongside `action_space` and `action_meanings`. Each experimental setup consisted of 10 independent runs per level with a maximum of 100 steps allowed per episode. Figure 4 presents the average win rates across our test suite, and most games showed a consistent 0% win rate across all models except for *Atlas Shrank* with a small probability of success and *Slidings* with a high probability for success for both ChatGPT 4o-mini and Deepseek-chat. In *Atlas Shrank*, this small nonzero win rate is likely owing to the first level being a simple tutorial level involving a relatively direct traversal of the map. In *Slidings*, the small number of movements needed to solve each level (with most levels requiring 4/5 movements to win) might have allowed the system to stumble upon correct solutions. This demonstrate difficulty in tracking interconnected rules and maintaining long-term plans, highlighting a significant gap between current LLM capabilities and the specialized problem-solving skills required for structured puzzle environments.

# 6 Discussion

Puzzle games present uncommon challenges for RL and LLM-based player agents. Specifically, efficient solutions require logical inference (e.g., deduction/induction) as well as long-range planning. Even apparently simple puzzle games can be fiendishly difficult in practice. This differs qualitatively from the challenges posed by video games such as first-person shooters or platform games; at the same time, these are single-player games, unlike classical board games such as Chess and Go. Another main issue with puzzle games is the late rewards, where the only reward is usually if you win. This sparsity of reward might pose a challenge for RL agents. This challenge might be harder in puzzle games than in other ones with sparse rewards due to the existence of deadlock states (states where the game is still playable but not winnable after reaching them). This might pose a great challenge even for curiosity-driven agents and other techniques used to battle sparsity.

To avoid overfitting or over-tailoring a method to a game, it is crucial to test on a number of games, preferably a large number. *PuzzleScript* fills that need, and *PuzzleJAX* makes it fast brings it into the modern deep learning ecosystem. The results highlight the difficulty of puzzle games in general, and offer a challenge to learning based methods—both those based on reinforcement learning and on large language models—as the only methods that are successful on multiple games are based on tree search. Solving the games as a human would solve them, without excessive testing of states by taking actions more or less blindly, is very much an unsolved challenge.

Crucially, as *PuzzleScript* is a generative description language rather than just a collection of games, this opens the door to automated or partially automated design of puzzle games. This could take the form of an AI-assisted game design tool, and/or an open-ended system which combines models learning to play games with another model learning to design them, in an evolutionary loop.

**Limitations.** Though most of the major features of *PuzzleScript* are replicated in *PuzzleJAX*, we identify in our dataset of human games many edge cases which are incompatible with our engine, either by violating our definition of the *PuzzleScript* DSL as a context-free grammar, or causing compile or runtime issues in our JAX environment, which have yet to be addressed. At the same time, having been designed with fidelity as a first priority, further speed optimizations are almost certainly possible. Meanwhile, we apply only simple, off-the-shelf algorithms to our domain in this preliminary study (foregoing, e.g. reasoning LLMs, which might have demonstrated enhanced performance on these complex puzzle-solving tasks).

# 7 Conclusion

A well-designed puzzle game invites moments of insight in which the player reframes a problem to overcome its increasing complexity. Our framework, *PuzzleJAX*, seeks to surface a space of problems in which apparent functional simplicity is juxtaposed with the surprising depth of thought required to arrive at a solution. By reimplementing *PuzzleScript*, an accessible and expressive game engine and Description Language with an active community of casual and professional users and designers, we not only gives AI researchers the ability to evaluate agents on hundreds of often carefully designed human games, but also provide a concise and expressive means of defining new novel problems. *PuzzleJAX* runs fast on the GPU by expressing rewrite rules as convolutional operations in Python's JAX library, and is by the same token easily connected to existing deep learning pipelines, while all the while remaining interoperable with *PuzzleScript*.

In preliminary testing, we find that naive breadth-first tree search does surprisingly well on a large number of games. Reinforcement Learning can quickly fall victim to local minima representing greedy strategies, and Large Language Models often become helplessly stuck in environments involving unconventional mechanics. This suggests the need for augmenting learning based methods with "insights" derived from search to produce more generally capable AI. *PuzzleJAX* provides a robust and efficient testing ground for such methods, in addition to other learning-based approaches focusing on exploration. One possibility is that general agents can only emerge via continual learning in a shifting landscape of semantically rich and varied tasks. *PuzzleJAX* makes such explorations possible via its concise description language, and may ultimately serve both as a benchmark for competent game-*playing* agents, and creative game *designing* agents.

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
