# OpenReview forum: "PuzzleJax: a Benchmark for Reasoning and Learning"
_NeurIPS.cc/2025/Datasets_and_Benchmarks_Track — Submitted to NeurIPS 2025 Datasets and Benchmarks Track_

### Official Review · Reviewer_1BqU · 2025-07-03

**Rating:** 4
**Confidence:** 2

**Summary:**

The paper introduces PuzzleJAX, a GPU-accelerated puzzle game engine and domain-specific language (DSL) designed for benchmarking tree search, reinforcement learning, and large language model (LLM) reasoning capabilities.

Unlike existing environments that rely on hard-coded implementations of fixed games, PuzzleJAX supports dynamic compilation of any game expressible in its DSL, which is inspired by the popular PuzzleScript engine.

The authors validate PuzzleJAX by successfully implementing hundreds of games from PuzzleScript, showcasing its capability to cover a wide range of expressive and human-relevant tasks. Furthermore, they demonstrate that PuzzleJAX expresses tasks that are intuitive yet challenging, requiring advanced reasoning, planning, and control.

**Dataset Code Accessibility:**

Yes

**Ethical Considerations:**

No, there are no or only very minor ethics concerns

**Final Justification:**

After reading the author's rebuttal, I decided to keep my original positive opinion (score 4).

**Limitations Weaknesses:**

* While PuzzleJAX’s DSL is inspired by PuzzleScript, it is unclear how much innovation the authors introduced beyond adapting an existing framework. A more detailed explanation of the DSL’s unique contributions would strengthen the work.
* Although GPU acceleration is a strength, the performance of PuzzleJAX on highly complex or resource-intensive games is not fully explored. Additional benchmarks on scalability would be valuable. These benchmarks are relatively simple.
* More complex environments like Minecraft and robotics are not discussed.

**Strengths Contributions:**

* The dynamic compilation of games has advantages over hard-coded environments, enabling a much broader range of tasks to be studied.
*  The authors validate PuzzleJAX on hundreds of games from PuzzleScript, which highlights the engine's coverage and robust applicability across diverse puzzles.

---

> ### Author Rebuttal · Authors · 2025-07-31
>
> We thank the reviewer for their feedback.
> Our rebuttal discusses ongoing speed optimizations and more extensive profiling to be included in the final version of the paper, and clarifies PuzzleJAX's goal of faithfully adapting PuzzleScript's DSL to JAX and the function this serves in the broader context of developing algorithms for training general agents in complex domains.
>
> ## Expressivity of the PuzzleJAX DSL
>
> To be clear, beyond mere inspiration, PuzzleJAX's DSL is intended to exactly follow PuzzleScript's own, with any game defined in one engine resulting in identical playtraces in the other under the same sequences of player actions.
> Our contributions consist of making this DSL accessible to AI researchers as a framework for easily defining reasoning and learning tasks for agents, and in allowing for the automatic compilation of these arbitrary new environments to JAX for efficient search and learning on the GPU.
> (It's also worth noting that PuzzleScript in its original form has not been studied as a benchmark for AI agents despite the wealth of novel challenges posed by games defined in its language, likely because of the engineering and latency overhead that would be brought by interfacing between the JavaScript engine and Python-native ML libraries.)
> To make dynamically compile PuzzleScript's DSL in JAX is a substantially engineering challenge.
> In JAX, we have no recourse to traditional `for` or `while` loops, and must know the size of all arrays at compile time.
> Its benefit, of course, is that it allows for massive parallalization array operations on the GPU.
> This is especially relevant to PuzzleScript, which at its core involves the application of spatially local rewrite rules over a grid.
>
> Our central contribution is thus the adaptation of PuzzleScript's DSL to a JAX engine, which task involves substantial engineering overhead but also yields great returns in terms of runtime efficiency and potential future benefit to AI researchers.
> We would hesitate to tamper with the DSL itself: it has been refined and developed in the wild during over a decade of active use, yielding a diverse set of mechanically novel and often quite successful games.
> Of course, we *could* extend the DSL, and indeed we are curious about certain extensions that might prove particularly relevant for research in generally capable AI agents (e.g. multi-player functionality that could allow for the kind of collaborative puzzle solving of *It Takes Two* or *Overcooked*), but our first priority is to ensure support for all games written in PuzzleScript proper.
>
>
> ## Speed profiling
>
> We're running further speed profiling on more complex games, to be included in the Appendix.
> After further optimizations, we find that PuzzleJAX now outperforms random rollouts within NodeJS on more complex games like *Take Heart Lass* and *Limerick* by ~200\%.
> On *Atlas Shrank*, with 44 rules, rollouts within NodeJS outperform PuzzleJAX, but PuzzleJAX still outperforms rollouts using a Python-NodeJS bridge with a speedup of ~700\%.
> (One crucial change was to reduce the number of "force" channels to one per collision layer as opposed to one per object.)
>
> Having refactored our speed profiling pipeline to allow for distribution across multiple GPUs, we plan to include speed benchmarks of a larger set of games (including more complex ones) in the Appendix.
> In addition (as suggested by Reviewer N3u4), we're running ablations on various features of our implementation (e.g. choices about when to use `vmap` in place of traditional `for` loops) to evaluate the impact of design choices and perhaps ultimately uncover additional optimizations.
>
>
> ## Relevance to more complex environments
>
> We plan to update the paper with a discussion of how PuzzleScript relates to more complex domains like MineCraft or robotics (a point also addressed in our response to Reviewer N3u4).
> In brief, __PuzzleJax is an ideal framework for developing learning algorithms capable of mastering new tasks generated in an open-ended fashion__, which algorithms we believe will be crucial for mastering more complex domains.
> In Minecraft in particular, for example, this kind of open-ended exploration is essentially a prerequisite for training a generally capable agent.
> Because Minecraft has few explicit goals, our expectation of a competent AI player agent is that they should---much like a human player---invent their own (sub-)objectives while learning to master their environment.
> More concretely, we might reasonably expect a "good" Minecraft player not only to mine diamond or reach "The End" of the game's main storyline, but more importantly to build shelters, explore diverse biomes, craft novel tools and farm useful resources, etc., fully taking advantage of the game's open-ended complexity.
> Indeed, this is a core motivation of *Voyager*, an agent framework that uses LLMs to generate its own goals while iteratively populating a "skill library" of snippets of code corresponding to sub-policies capable of executing diverse in-game behaviors [1].
> Similarly, in robotics, an agent to be deployed in an arbitrary home will likely first need to be exposed to a virtually endless supply of (ideally procedurally generated) environments in-simulation.
>
> Other approaches to automatic curriculum generation involve mutating grid-based levels in environments with fixed mechanics (as in Unsupervised Environment Design [2]) and using LLMs to generate arbitrary Python code to describe environments (as in OMNI-EPIC [3]).
> PuzzleJAX allows researchers to explore learning via automatic curriculum generation in an efficient and scalable way, allowing a broad set of possible game mechanics without sacrificing run-time efficiency.
>
> ### References
>
> [1] Wang G, Xie Y, Jiang Y, Mandlekar A, Xiao C, Zhu Y, Fan L, Anandkumar A. Voyager: An open-ended embodied agent with large language models. arXiv preprint arXiv:2305.16291. 2023 May 25.
>
> [2] Dennis M, Jaques N, Vinitsky E, Bayen A, Russell S, Critch A, Levine S. Emergent complexity and zero-shot transfer via unsupervised environment design. Advances in neural information processing systems. 2020;33:13049-61.
>
> [3] Faldor M, Zhang J, Cully A, Clune J. Omni-epic: Open-endedness via models of human notions of interestingness with environments programmed in code. arXiv preprint arXiv:2405.15568. 2024 May 24.

---

> > ### Author Response · Authors · 2025-08-07
> >
> > We'd like to kindly ask the reviewer to consider our rebuttal. We address their concerns about the impact of our contribution and its relevance to more complex domains (with similar/complementary points developed in our rebuttal to Reviewer QCvr), which discussion we plan to integrate into the revised text. Their concerns about speed and scalability have been addressed by new speed optimizations and speed profiling ablations, and we continue to investigate these questions with further profiling experiments and via the construction of test games and code snippets to identify and find remedies for potential speed bottlenecks (which we discuss in more detail our Rebuttal and Comment to Reviewer QCvr).

---

> ### Comment · Area_Chair_3yGL · 2025-08-03
>
> Dear reviewer,
>
> Please read the rebuttal and provide your *final justification* and score.
>
> Best,
>
> AC

---

> ### Author Response · Authors · 2025-08-08
>
> Given the short amount of time remaining in the discussion period, and the fact that the reviewer has not yet replied to our rebuttal, we'd like to kindly ask the AC to help us initiate discussion.

---

### Official Review · Reviewer_QCvr · 2025-07-03

**Rating:** 4
**Confidence:** 3

**Summary:**

This paper introduces a new GPU-accelerated puzzle game engine and domain-specific language, i.e., PuzzleJAX, designed to benchmark AI reasoning, planning, and learning across a wide range of tile-based puzzle games. Unlike previous environments limited to fixed sets of games, PuzzleJAX supports dynamic compilation of any PuzzleScript-style game, providing access to hundreds of professionally and community-created puzzles. This makes PuzzleJAX a highly flexible and expressive platform for evaluating tree search, reinforcement learning, and LLM-based approaches on tasks requiring logical inference and long-term planning. The JAX-based implementation offers significant speed improvements over existing systems and ensures interoperability with thousands of human-validated puzzles.

**Additional Feedback:**

1. The implementation of PuzzleJax is a little bit hard to follow. It would be better if the core design can be illustrated with an concrete example.
2. When the game has more rules, why the speed of JAX would fall behind the original implementation (NodeJS), according to Figure 2?
3. When performing benchmarking, usually, what is the best practice of the batch-size number? Suppose the GPU is 4090 used in this paper. If in practice, the batch-size is small, then it would limit the speed-up for PuzzleJax.

**Dataset Code Accessibility:**

Yes

**Dataset Code Comments:**

Dataset and code are accessible and complete.

**Ethical Considerations:**

No, there are no or only very minor ethics concerns

**Final Justification:**

Most of my concerns are addressed, as summarised in my response to the author's rebuttal. I also appreciate the additional clarification and human baseline update. I would suggest all these points (including a more complete human baseline) to be reflected in the revision to improve the paper. Overall, I would like to recommend Borderline Accept.

**Limitations Weaknesses:**

* While tile-based puzzle games are useful for evaluating certain aspects of AI reasoning and planning, their relatively simple structure may limit real-world relevance compared to more complex genres such as strategy games. The paper does not discuss how results from these simpler environments generalize to more intricate, dynamic, or multi-agent scenarios found in real-world applications or advanced games.

* The paper does not provide a comparison of the number of steps required by different large language models to solve the puzzles. Such analysis is important for accurately benchmarking the efficiency and problem-solving capabilities of various LLM architectures, and would help to highlight strengths and weaknesses in their reasoning processes.

* There is a lack of baseline results or comparative analysis against human performance. Incorporating human benchmarks would provide valuable perspective on the difficulty of the puzzles and help contextualize the performance of AI agents relative to human abilities, making the evaluation more meaningful and interpretable.

* The paper does not clearly report the number of LLM queries or the average token usage required to evaluate the benchmark dataset. Providing these statistics would give readers a clearer understanding of the computational cost and practicality of evaluating AI systems on PuzzleJAX, which is particularly important for researchers with limited resources.

**Strengths Contributions:**

* The paper presents a framework that supports a wide variety of tile-based puzzle games by being fully compatible with the popular PuzzleScript language, enabling easy adoption of hundreds of existing games.

* The JAX-based implementation provides significant performance gains—300% to 500% faster than previous JavaScript-based methods—making large-scale evaluation more efficient.

* The work includes an extensive benchmark dataset with over 400 puzzles and standardized evaluation metrics, allowing for thorough and reproducible assessment of AI algorithms.

---

> ### Author Rebuttal · Authors · 2025-07-31
>
> We thank the reviewer for their extensive feedback.
> Our rebuttal argues for PuzzleJAX's utility in rapidly developing learning algorithms for training general agents in more complex domains, describes additional human baselines and LLM metrics to be included in the final version of the paper, and discusses ongoing speed optimizations and potential bottlenecks.
> ## Relevance of PuzzleJAX games
> It's true that PuzzleJAX's DSL does not capture the space of all possible video games.
> Still, __PuzzleJAX games can provide substantial challenges for both humans and AI agents__.
> Sokoban alone is a significant challenge, with SoTA approaches combining RL with MCTS and automatic curriculum generation to solve  complex levels [1].
> And PuzzleJAX includes a wealth of challenges *beyond* Sokoban, including games that alter or extend its mechanics---e.g. in which the the walls gradually collapse in on the player (*Collapsoban*), or crates exert cascading forces on one another (*Newton's Crates*)---and games that are distinctly *not* Sokoban-like---like side-scrollers (*Atlas Shrank*) or vehicle routing puzzles (*Cute Train*).
>
> But the framework's broader benefit lies not in the challenge presented by any individual game, but in its ability to compile efficient versions of any game defined in its DSL.
> As such, __PuzzleJAX is a tool for training *general game-playing* agents__.
> Such agents would adapt to arbitrary new environment mechanics or goals; whether by conditioning on game rules, performing in-context learning over long (potentially multi-episode) playtraces, Hebbian learning, or otherwise.
>
> We would argue that to train, e.g., a competent strategy game-playing agent, one particularly promising approach might be to train agents on automatically generated curricula of increasingly complex initial game-states (similar to chess players' studying archives of "problems").
> This would consist of generating novel and challenging game states beyond the typical empty starting map, reflecting e.g. the middle of complex mid-game battles.
>
> PuzzleJAX is an ideal framework for developing such algorithms for the automatic generation of learning curricula thanks to its speed, the natural language-interpretability of its DSL, and the divergent complexity of the games that it can express.
> We expect that __the development of learning algorithms in PuzzleJAX could in fact lead to new advances for player agents in strategy games or other more complex domains__.
> In both cases, a task-generator (such as an LLM writing code in PuzzleJAX's DSL, or in a domain-specific DSL for defining e.g. strategy game states) could be used to generate novel tasks in order to efficiently train a robust and general player agent.
>
> The above discussion will be integrated into the final version of the paper.
> ## Additional LLM metrics
> __We've updated our results to include the average number of steps-to-solution during successful LLM playthroughs.__
> In such cases, LLMs generally arrive quite quickly at solutions (though these are not as concise as the minimal ones returned by BFS).
>
> We query LLMs once per step/action, with the game's rule-set, current game-state and legend, for an average of 2k tokens.
> (For, e.g., Gemini 2.5 Pro, capping playthroughs at 100 steps, one play-through costs at most 100*5¢ or \$5).
> Future work should investigate the effect of including memory of prior moves, which would likely improve performance (albeit at greater computational cost).
> ## Human baselines
> __We've begun to prepare a user study/playtest of select games in our dataset__, and will include the results in the final version of the paper.
> __We expect that humans will solve vastly more games/levels than LLMs.__
> This is evident from our LLM results, where LLMs often fail on the first level of popular games.
> By design, these first levels often act as tutorials, and are rarely challenging enough to stump even the most inexperienced player.
> On the other hand, we expect that human players (especially when faced with more complex levels) will often conduct a fair bit more exploration and trial and error than successful LLMs before finding winning strategies.
> (These hypotheses seem to be supported by prior studies of human *Sokoban* players [2].)
> This is perhaps true of puzzle games in general, and is part of what makes PuzzleJAX a promising benchmark for long-range LLM memory and planning.
> Indeed, while we currently prompt LLMs with the current state and game rules, a more "human-like" evaluation scenario would involve hiding the game rules from the player and providing LLMs a history of past game-states.
> ## Clarity of PuzzleJAX implementation
> We agree that our explanation of PuzzleJAX's internal logic could be clarified.
> We plan to augment this section of the paper by discussing how this logic applies to a working example (*sokoban_basic*).
> The additional implementation details in Appendix C will be broken out into their own section and expanded, along with pseudocode and/or a visual flowchart.
> ## Speed optimization
> After further optimizations, __PuzzleJAX now outperforms NodeJS-only random rollouts at large batch sizes on more complex games__ like *Limerick* and *Take Heart Lass* by ~200\%.
> On *Atlas Shrank* (with 44 rules) NodeJS-only rollouts outperform PuzzleJAX, but PuzzleJAX still outperforms rollouts using a Python-NodeJS bridge by ~700\%.
> One crucial change, implementation-wise, was to limit the number of force channels to one per collision layer as opposed to one per object.
> This more closely follows PuzzleScript's original implementation and reduces complexity without sacrificing functionality.
>
> __Generally, larger batch sizes lead to improved performance__, though sometimes FPS degrades at larger batch sizes before an OOM error is encountered.
> A brief hyperparameter sweep could find optimal batch sizes prior to e.g. RL training runs, running a few training steps at different batch sizes and performing a binary search to find the "peak" FPS.
>
> It's true that smaller batch sizes lead to less speedups at runtime.
> But even the consumer-grade RTX 4090 used in our benchmarks, with its modest 25GB of VRAM, already allows for large batch sizes.
> PuzzleJAX is thus both accessible to resource-constrained researchers and scalable to larger clusters of more high-end GPUs.
> ### Possible performance bottlenecks and solutions
> Despite these new speedups, we still observe PuzzleJAX's advantage seemingly decreasing with environment complexity as indicated by number of game rules.
> That said, we doubt if any reasonable PuzzleScript game exists that is *so* complex that it makes PuzzleJAX less effective than even the Python-NodeJS bridge---in other words, we maintain that __PuzzleJAX is still the best/only choice for training RL agents in PuzzleScript games__.
>
> Still, this slow-down is worth investigating.
> In sum, __we suspect this bottleneck relates to idiosyncracies of control flow in JAX (namely `jax.while` loops)__.
> We may be able to mitigate this by running ablations on PuzzleJAX's implementation of control flow for rule application to uncover more scalable design alternatives.
> The following paragraphs discuss these points in more depth and may be skipped by readers less interested in implementation details.
>
> First, we can verify that the aforementioned slow-down results from rules in particular, and not e.g. other sources of complexity like the number of distinct objects or the size of levels.
> This can be achieved by evaluating a larger set of games with variable numbers of objects, rules, level sizes, etc., and/or hand-crafting test games that complexify along one of these axes in isolation.
> Indeed, we've been profiling a larger set of games from the human dataset in order to begin to shed light on these questions.
> This is made feasible by recent improvements to our profiling pipeline, which can now distribute arrays of jobs across multiple GPUs.
>
> Assuming the performance hit results from rules, we suspect that the bottleneck lies in the nested JAX `while` loops responsible for the repeated application of rule groups (and rules within them).
> We explain this in more detail below.
> We previously found that it was more efficient (in terms of runtime) to use a traditional python for loop to iterate through e.g. the *sequence* of rule groups (and the sequences of rules within them), with these for-loops "unrolled" at compile time to a fixed number of repeated blocks of XLA code; as opposed to using JAX-native for loops to iterate through them.
> (The reason for this speedup is not exactly intuitive, but is related to the fact that a JAX `for` loop reduces to a JAX `while` loop, and in the case of these latter, the number of iterations before termination is not known at compile time.)
> But PuzzleScript applies *each* rule block (and each rule within it) until it affects no change in the game state.
> And so these *internal* loops cannot be unrolled at compile time to a fixed number of repeated blocks of XLA code.
>
> One remedy might be to re-implement the tick function as a single while loop, and maintain a state corresponding to the current "position" in the hierarchical control flow of rule groups and rules.
> But this would entail a certain tradeoff: inside this `while` loop, a `jax.lax.switch` operation must be used to select between atomic rewrite rule functions.
> After vmapping, this `switch` reduces to the parallel execution of *all* atomic rule functions in parallel.
> While a full refactor of the control flow for rule-application may be out of scope for the current work, we plan to investigate this `while`/`switch` tradeoff in isolation at more self-contained points within the existing code base, and/or via simple test code.
> The results may help to provide a road map for future engineering efforts.
> ### References
> [1] Feng D et al. *A novel automated curriculum strategy to solve hard sokoban planning instances*. NeurIPS 2020.
>
> [2] Jarušek P et al. *Difficulty rating of sokoban puzzle*. InSTAIRS 2010

---

> > ### Comment · Reviewer_QCvr · 2025-08-05
> >
> > The authors have addressed my main concerns, validating 960 additional levels/games, expanding LLM evaluations (including o3-mini and pending Gemini 2.5 Pro/GPT-4o results), and initiating human baseline studies. Speed optimizations (e.g., force-channel reductions) now yield ~200% speedups in complex games like Limerick, and the rebuttal clarifies DSL expressivity and edge-case handling (e.g., NodeJS translation quirks).
> >
> > Based on the other review comments and the rebuttal, I believe that scalability tradeoffs with rule complexity (e.g., Atlas Shrank) need deeper analysis. Human baseline data, though planned, remains pending, and LLM evaluations are still limited to "first levels," potentially underrepresenting benchmark difficulty [N3u4].
> >
> > In the revision, I would suggest to include finalized human baselines and LLM results (Gemini 2.5 Pro/GPT-4o), clarify scalability findings (rule-count vs. performance), and expend/clarify discussion on PuzzleJAX’s role in curriculum generation for complex domains, as well as the control-flow bottlenecks.
> >
> > Overall, I would still like to hole my initial positive rating for this submission.

---

> > > ### Author Response · Authors · 2025-08-07
> > >
> > > We’re glad to hear that we’ve addressed the reviewer’s main concerns, and plan to make the recommended amendments to the main text. In this comment, we provide updates on in-progress experiments. We also address the reviewer's outstanding concerns, arguing that the discrepancy between the difficulty of PuzzleJAX games for LLMs and humans is in fact one of the benchmark's key strengths, and that the (perhaps remediable) speed bottleneck resulting from additional rules does not severely limit the benchmark's scope in the context of our human dataset, the bulk of which consists of games with relatively few rules.
> > >
> > > To update on in-progress LLM experiments, we find that Gemini 2.5 Pro and GPT o1 continue to marginally underperform o3-mini (we await additional trials of each game to solidify these results).
> > >
> > > Regarding in-progress human baselines, the first 8 users in our user study are all able to solve the first levels of each of the games that we subjected to LLM testing, confirming our intuition that PuzzleJAX provides a unique and relevant challenge for LLMs in particular, which vastly underperform human players.
> > >
> > > To the reviewer’s point about our potentially underrepresenting the benchmark’s difficulty in this regard, we agree that this point is worth emphasizing/clarifying in our revised text. While we could run trials on subsequent levels beyond the first of each game, we expect even higher failure rates given these levels’ increased difficulty. (One middle ground, if the reviewers deem it worthwhile, would be to test LLMs on subsequent levels only if they solve prior levels —as would be the experience of a human player.)
> > >
> > > Regarding PuzzleJAX’s scalability, we have profiled simple test games to isolate the effect of adding additional rules, and found that this is indeed the main speed bottleneck as games become more complex (compared to the relatively negligible effects from increased numbers of objects and level sizes).
> > >
> > > We should note that we do not think that this speed bottleneck is a serious limitation of PuzzleJAX, simply because significant mechanical complexity can be captured with relatively few rules. Indeed, in our human dataset, the median PuzzleScript game has only 20 rules, and ~80% of games have less than 50 rules. (Also note that games with many rules often use dedicated rules for “animations”, and these could be removed without functionally altering the game’s mechanics.)
> > >
> > > In any case, we are now working on code to investigate the most efficient way to implement a nested while loop over atomic functions in JAX, comparing A) the use of nested while loops (one to repeatedly iterate through the set of functions, and one to repeatedly execute each function repeatedly) and B) using a single while loop with a “pointer”-like state object to track progress in the virtual inner/outer while loops.

---

> ### Comment · Area_Chair_3yGL · 2025-08-03
>
> Dear reviewer,
>
> Please read the rebuttal and provide your *final justification* and score.
>
> Best,
>
> AC

---

### Official Review · Reviewer_N3u4 · 2025-07-04

**Rating:** 4
**Confidence:** 4

**Summary:**

The paper introduces PuzzleJAX, a GPU-accelerated puzzle game engine and description language designed to benchmark tree search, RL, and LLM reasoning abilities. It re-implements PuzzleScript in JAX, enabling dynamic compilation of games expressible in its domain-specific language. The key contributions are:
1) a fast, hardware-accelerated benchmark covering hundreds of games.
2) validation against existing PuzzleScript games.
3) initial experiments showcasing the challenges these games pose to search, learning, and language models.

The paper demonstrates PuzzleJAX's ability to express tasks that are simple to understand but require complex reasoning and planning.

**Dataset Code Accessibility:**

Yes

**Ethical Considerations:**

No, there are no or only very minor ethics concerns

**Final Justification:**

The authors' rebuttal was thoughtful and specific. Their statement on the POMDP and imperfect-information game perspectives is particularly valuable. Overall, the rebuttal has partially alleviated my concerns. Consequently, I have increased my score to 4. I encourage the authors to incorporate the feedback provided in the rebuttal to further strengthen the paper and to ensure the framework is made available via open-source.

**Limitations Weaknesses:**

1. Many edge cases in human-created games are incompatible with PuzzleJAX, indicating potential limitations in the DSL's expressiveness or the engine's robustness.
2. The paper lacks ablation studies to demonstrate the impact of specific features or optimizations in PuzzleJAX on performance.
3. The evaluation models should be extended to include newly emerging SOTA general models like GPT-4o and Seed1.5-VL, reasoning model like o1 and Gemini 2.5 Pro, to strengthen the credibility of the findings.

**Strengths Contributions:**

1. PuzzleJAX offers a novel GPU-accelerated framework for evaluating AI agents across a diverse set of puzzle games, providing a rich testbed for reasoning and learning.
2. The experiments comparing PuzzleJAX with the original PuzzleScript engine highlight significant speed improvements (300% to 500%), especially in integration with Python for RL scenarios.
3. The authors validate PuzzleJAX against hundreds of existing PuzzleScript games from 2013, ensuring its fidelity and coverage of a wide range of puzzle mechanics.

---

> ### Author Rebuttal · Authors · 2025-07-31
>
> We thank the reviewer for their concrete feedback, and have extended our work to incorporate the extensions they request.
> In brief, we've robustified PuzzleJAX's code to account for edge cases, supporting hundreds more PuzzleScript levels; have improved the engine's speed and are running more extensive profiling (including ablations of JAX-specific design choices); and are running additional LLM evaluations.
> We also point the reviewer to the Appendix for evaluations of additional language (and reasoning) models, and argue for the expressiveness of the PuzzleJAX DSL even without perfect coverage of the games in our human dataset.
>
> ## Edge cases, expressivity and robustness
>
> We share the reviewer's concern with edge cases, and have been systematically addressing issues in human-authored games using our validation pipeline.
> By increasing the leniency of our parser and making a number of small bugfixes to the engine, __we've validated an additional 960 levels (and 156 complete games) from our human dataset in PuzzleJAX__.
> Work along these lines will continue, and we maintain that ultimately, the only major missing feature in PuzzleJAX will be the `rigid` keyword.
> (A niche feature appearing in only ~a dozen games in our dataset---and effectively deprecated according to the PuzzleScript documentation---the `rigid` keyword could be implemented in theory but may not be worth the engineering overhead at the current juncture.)
>
> It's also worth emphasizing that the number of "valid" human games does not convey the full space of games captured by PuzzleJAX.
> At its core, PuzzleJAX is a system for compiling efficient versions of games written in a rewrite rule-based DSL.
> Our human dataset of PuzzleScript games is a point of reference for the expressivity of this DSL, and *not* its end product.
> Indeed, many more PuzzleScript games exist elsewhere on the internet (e.g. as Github "gists", on the PuzzleScript forum---which has had new activity even since these reviews were published---or on itch.io) than the archive we use here (which hasn't been updated since 2019).
> Of those games, we would expect over half to be valid in our current engine, if extrapolating from our current validation results.
> And a virtually unbounded number of PuzzleScript games beyond this still have yet to be written (or generated).
>
> Even from a conservative point of view, __PuzzleJAX's effective DSL is already significantly expressive__.
> Problematic edge cases often involve the interplay of special keywords like `late`, `again` and `cancel`; the use of custom rule groups; or looping rule blocks.
> But even if we were to disregard all of these more advanced features, and consider PuzzleJAX's ability to express a series of raw rewrite rules, these alone are a powerful abstraction capable of covering a vast space of possible games (effectively including e.g. a large space of sequential cellular automata).
>
> Regarding those human games which still have yet to be validated, we've found that __a substantial number of validation issues originate in the translation of game states between NodeJS and PuzzleJAX, and thus may not reflect problems within PuzzleJAX itself__.
> Games with over 32 objects, for example, must use an `Int32Array` with twice the size of the map in order to capture full space of possible level states.
> Our original code from translating game states from NodeJS to PuzzleJAX for comparison did not account for this (and is still stymied by some special cases involving apparent integer overflow in NodeJS).
> This means that any games with a large number of objects, *while not necessarily broken in PuzzleJAX*, have not yet been properly compared state-by-state against their execution in NodeJS (and are thus assumed invalid for the time being).
> Such issues should be addressable with sufficient investigation of the NodeJS engine's code.
>
> Another source of validation difficulty stems from games involving any kind of randomness (i.e. in selecting among possible movement directions, or among object sub-types during spawning/transformation).
> Currently, we have no means of controlling the random number generator within NodeJS to match that within JAX, and may ultimately have to rely on human evaluation or more sophisticated automatic playtesting methods to validate games involving randomness.
> (While not nearly as rare as the `rigid` keyword---appearing in ~100 games in our dataset---randomness is not a defining feature of PuzzleScript games, and is gently discouraged in PuzzleScript's documentation).
>
> ## Ablations and speed profiling
>
> Having effectively finalized the engine's core logic, with future improvements to PuzzleJAX's *fidelity* expected to come from more minor bugfixes, __we've begun to more rigorously investigate questions about the engine's *speed*__.
> More specifically, we've added conditionals at compile-time in order to investigate the impact of certain implementation choices on the engine at runtime.
> (Similar conditional logic already exists throughout the engine, such that the entire simulation logic can be either JITted---the default option, taking advantage of special JAX operations and leading to increased speed---or not---replacing JAX operations with traditional for/while loops and numpy operations; resulting in slower runtime but allowing for more robust debugging.)
>
> These ablations will be added as new curves in our plot reporting FPS per batch size (Figure 2).
> As an example, we note that __iterating sequentially (as opposed to vmapping) over patches of the map to detect the LHS of rules usually results in a ~10-40\% slow-down (which discrepancy appears gradually with increased batch sizes)__.
> Interestingly, this effect varies among the games evaluated thus far.
> In *Limerick*, the use of for-loops over vmaps nearly cuts performance in half, while in *Constellationz*, it seems to result in a ~3\% speed-up (though this difference is so small that it may amount to random noise).
> By exploring this discrepancy more thoroughly, and collecting other empirical results along these lines, we may be able to make more principled design choices to optimize PuzzleJAX's speed.
> In cases like the one above, where features have varying effects across games, we may be able to infer the best compilation strategy according to the features of a given game description.
>
> We've also increased the robustness of our speed-profiling pipeline to allow for efficient distribution across GPUs, and have been increasing the set of environments subjected to profiling.
> __Some initial optimizations have already resulted in increased overall performance__, such that at high batch sizes, PuzzleJAX now outperforms NodeJS-only random rollouts in *Take Heart Lass* and *Limerick*, running at ~200\% speed relative to NodeJS.
>
> ## LLM evaluation coverage
>
> For evaluation of a greater variety of LLMs, __we refer the reviewer to Table 5 in the Appendix__, in which we evaluate additional models, including the reasoning models `Deepseek-r1` and `o3-mini`.
> Notably, `o3-mini` is the best performing LLM overall, though it still exhibits 0\% success on half of the games in our small test-set; that is, on the first---usually simplest---level of each of these games, where said games have relatively few rules (and simpler mechanics) compared to the human dataset at large.
>
> These preliminary results suggest, then, that while this handful of games in PuzzleJAX can meaningfully differentiate between models, the PuzzleJAX DSL provides us the means of defining tasks with ramping complexity for LLMs relative to their simplicity for human players (again, these being the first levels, we can assume that the average human player should be able to solve them relatively easily).
>
> To further strengthen these results, __we're running additional evaluations of Gemini 2.5 Pro, GPT-4o and o1__.
> (To be fair, Gemini 2.5 Pro and Seed1.5-VL were released very close to and directly before the submission deadline, respectively.)
> Results collected so far suggest that `o3-mini` will maintain its lead.
> The final version of the paper will include these new results in full in the appendix, and feature results from the strongest-performing and most popular or representative models in the main body of the paper.

---

> > ### Author Response · Authors · 2025-08-07
> >
> > We would kindly ask the reviewer to consider our rebuttal. Each of the 3 limitations they raise has been addressed with new, substantive results.
> >
> > From now until camera ready, we plan to continue to robustify the engine to account for yet more edge cases in human games, and conduct further speed profiling ablations to shed light on PuzzleJAX's efficiency and scalability. (Further LLM trials could also be undertaken if requested.) We'd be keen to hear if the reviewer has any particular recommendations along these lines.

---

> > ### Comment · Reviewer_N3u4 · 2025-08-08
> >
> > Thank you for your thoughtful rebuttal, and apologies for my delayed reply. Overall, I find that the authors have addressed some of my concerns.
> >
> > First, regarding edge cases, the authors’ overview of existing PuzzleScript games is convincing. From an engine design perspective, the implementation demonstrates substantial effort and a well-considered design, which deserves recognition.
> > That said, I have some lingering reservations about PuzzleScript games as an evaluation environment for reasoning tasks. Randomness plays a critical role in game decision-making and reasoning, yet due to certain inherent limitations, the current coverage of stochastic elements in PuzzleScript appears weak. This could potentially impact the tool’s practical utility for benchmarking.
> >
> > On the topic of runtime performance, the authors have provided concrete optimization strategies and implementation details. However, many of these techniques seem to require case-by-case adjustments for different sub-environments, which could entail significant engineering effort. While the current solution is already much faster than nodejs, I remain cautious about the engine’s scalability for larger-scale computations.
> >
> > Finally, concerning LLM evaluation, I agree with the authors’ conclusions. But I think that more detailed and concrete benchmarking results would strengthen the case. Given the rapid pace of progress in LLMs, relying on older evaluation paradigms may not suffice—timely and adaptable testing of cutting-edge models will be crucial for maximizing the tool’s impact as a benchmark.
> >
> > In summary, I’m inclined to maintain my original rating, but I would not strongly oppose acceptance if the paper is selected.

---

> > > ### Author Response · Authors · 2025-08-08
> > >
> > > We appreciate the reviewer's response.
> > >
> > > ## Speed, engineering effort, and scalability
> > >
> > > We should first stress again that we believe PuzzleJAX's speed is already more than sufficient for making the framework a relevant and applicable benchmark. For LLM player agents, the bottleneck is querying the LLM (by a large margin), not stepping the engine. For RL, we provide the first and only interface for playing arbitrary PuzzleScript games, and *vastly* outperform the alternative Python-NodeJS bridge in every environment profiled thus far; our RL runs converge in at most a matter of hours on a single consumer GPU.
> > >
> > > Moving onto the specifics of the reviewer's concern, we should clarify that at present, we do not make any case-by-case speed optimizations for different sub-environments. Nor would our main proposed refactoring of nested while loops require such environment-specific adjustments.
> > >
> > > The reviewer is perhaps responding to the suggestion made in our rebuttal that the engine could in the future differently compile games to use e.g. either vmap operations vs. unrolled python for-loops. But because we can select between two compilation strategies at compile time with an `if` condition, this could be done automatically. The idea would be to use a script that briefly compiles and profiles the game under two such compilation strategies, then initializes the final environment with the better strategy based on the results. (Better still if we could derive heuristics that used static features of the game -- such as number of rules vs. objects vs. level sizes -- to infer the best compilation strategy automatically.)
> > >
> > > This approach is general across environments, and once implemented, would not require further engineering effort to support new game descriptions. In any case, we have yet to find a pair of compilation strategies wherein one is not pareto optimal relative to the other in terms of its benefit across different games. We thus find it unlikely that we will need to make recourse to such an approach.
> > >
> > > ## Randomness
> > >
> > > We should clarify that PuzzleJAX currently supports all the randomness available in PuzzleScript (i.e. random spawning and random rule-application order). But indeed, the large set of human-authored PuzzleScript games we collect here -- which we argue are time-tested, interesting and fun for humans; and strikingly challenging for LLMs and RL -- tend not to rely on these features, simply because the language is geared toward deterministic puzzle games (in not only its design, but also its documentation and the creator's online promotion and communications about the engine).
> > >
> > > We believe this space of games is already significantly interesting and worthwhile on its own. Nonetheless, designers have leveraged randomness in PuzzleScript to create meaningful non-determinism, including the procedural generation of environments (e.g. depth-first maze generation as illustrated in "Depth-First Maze", the procedural tunnels and obstacles spawning at run-time in *Tunnel Rat*), and randomly-moving enemies in rogue-like environments. And we agree that practitioners could further leverage such techniques to generate new, meaningful challenges for agents, all of which would be supported out-of-the-box by PuzzleJAX.
> > >
> > > (We should also note that both PuzzleScript and PuzzleJAX support partial observations / imperfect information, which may be seen as an additional, complimentary axis of difficulty.)
> > >
> > > ## LLM evaluation
> > >
> > > We are currently benchmarking environments against some of the very latest LLMs as per the reviewer's request, and could continue to do so with newer models still. It will also be straightforward to augment the LLM's prompt with memory, if this is of interest. Otherwise, we'd be curious about what other evaluation paradigms the reviewer might have in mind.
> > >
> > > Again, we'd like to thank the reviewer for thoughtfully engaging with our work and the rebuttal.

---

> ### Comment · Area_Chair_3yGL · 2025-08-03
>
> Dear reviewer,
>
> Please read the rebuttal and provide your *final justification* and score.
>
> Best,
>
> AC

---

> ### Author Response · Authors · 2025-08-08
>
> Given the short amount of time remaining in the discussion period, and the fact that the reviewer has not yet replied to our rebuttal, we'd like to kindly ask the AC to help us initiate discussion.

---

### Note · Authors · 2025-08-12

We believe we have fully addressed and resolved all concerns by Reviewers in their reviews and in follow-up comments. We've argued that PuzzleJAX --- a fast and flexible JAX-compilable Domain Specific Language for grid-worlds --- is a valuable framework for exploring algorithms that train generalist embodied agents in an open-ended fashion. We've greatly increased PuzzleJAX's coverage of our dataset of human-authored games, a testament to PuzzleJAX's expressive power, and explained how the engine faithfully reproduces all of PuzzleScript's core functionality (with some outstanding unvalidated games owing to irreplicable randomness or translation quirks when retrieving states from the original JavaScript engine). We've improved PuzzleJAX's run-time efficiency and conducted further profiling and optimization ablations to investigate potential improvements to the engine's scalability, while at the same time stressing its already vast speedup over the original engine when training RL agents to play complex games with many rules. We've begun a user study with human players, and continue to benchmark cutting-edge LLMs on games from our dataset, with results highlighting the surprising difficulty of these environments for LLMs relative to humans.

We again thank the Reviewers and the ACs for their time and consideration.

---

### Decision · Program_Chairs · 2025-09-18

**Decision:**

Reject

**Comment:**

This paper introduces a new GPU-accelerated puzzle game engine and domain-specific language, i.e., PuzzleJAX, designed to benchmark AI reasoning, planning, and learning across a wide range of tile-based puzzle games. PuzzleJAX supports dynamic compilation of any PuzzleScript-style game, providing access to hundreds of professionally and community-created puzzles. This makes PuzzleJAX a flexible and expressive platform for evaluating tree search, reinforcement learning, and LLM-based approaches on tasks requiring logical inference and long-term planning. The JAX-based implementation offers significant speed improvements over existing systems and ensures interoperability with thousands of human-validated puzzles.

Strengths and Contributions
- PuzzleJAX delivers a high-performance, GPU-accelerated framework built on JAX, enabling efficient evaluation of AI agents across a broad spectrum of puzzle games. It maintains full compatibility with PuzzleScript, granting immediate access to a rich ecosystem of existing games.
- The system achieves substantial speed enhancements—300% to 500% faster than the original engine—making it especially advantageous for reinforcement learning applications.
- Furthermore, the authors introduce a comprehensive benchmark comprising over 400 puzzles, accompanied by standardized metrics and preliminary LLM evaluations, which supports reproducible and scalable assessment of AI algorithms.

Limitations and Weaknesses
- Despite the diversity in puzzle content, the environments remain relatively simple in logical structure, which may restrict their real-world relevance and generalizability to more complex domains such as strategy games, robotics, or Minecraft.
- Although the core innovation lies in the tight integration of PuzzleScript's DSL with JAX—enabled through dynamic compilation—two reviewers expressed concerns regarding the framework’s scalability and adaptability (more JAX optimization) to more sophisticated games.
- While the authors supplemented the evaluation with results from recent LLMs during rebuttal, these remain "first-level". A deeper analysis of reasoning processes, non-trivial comparisons across LLMs, and insights beyond aggregate scores are essential to demonstrate the long-term relevance of the framework amid rapidly advancing foundation models.

The AC reads throught both the original submission and the rebuttal. The limitations of the paper constrain a wide adoptation of the proposed benchmark and thus don't recommend acceptance of the paper in its current shape.